# Everolimus Mitigates the Risk of Hepatocellular Carcinoma Recurrence after Liver Transplantation

**DOI:** 10.3390/cancers16071243

**Published:** 2024-03-22

**Authors:** Paolo De Simone, Arianna Precisi, Quirino Lai, Juri Ducci, Daniela Campani, Piero Marchetti, Stefano Gitto

**Affiliations:** 1Liver Transplant Program, University of Pisa Medical School Hospital, 56124 Pisa, Italy; 2Department of Surgical, Medical, Biochemical Pathology and Intensive Care, University of Pisa, 56126 Pisa, Italy; daniela.campani@unipi.it; 3Transplant Laboratory, University of Pisa Medical School Hospital, 56126 Pisa, Italy; a.precisi@ao-pisa.toscana.it; 4AOU Umberto I Policlinico of Rome, Sapienza University of Rome, 00161 Rome, Italy; quirino.lai@uniroma1.it; 5Azienda Ospedaliero Universitaria Pisana, 56124 Pisa, Italy; j.ducci@ao-pisa.toscana.it; 6Department of Pathology, University of Pisa Medical School Hospital, 56124 Pisa, Italy; 7Diabetology Unit, University of Pisa Medical School Hospital, 56124 Pisa, Italy; piero.marchetti@unipi.it; 8Internal Medicine and Liver Unit, University Hospital Careggi, 50134 Florence, Italy; stefano.gitto@unifi.it; 9Department of Experimental and Clinical Medicine, University of Florence, 50134 Florence, Italy

**Keywords:** liver transplantation, immunosuppression, everolimus, hepatocellular carcinoma, recurrence

## Abstract

**Simple Summary:**

Everolimus is an immunosuppressive drug used to prevent rejection after liver transplantation. It is an attractive alternative to tacrolimus for patients with hepatocellular carcinoma who are undergoing liver transplantation due to its antiproliferative effects. In our study, we investigated whether liver transplant patients who received everolimus after transplantation had a reduced risk of hepatocellular carcinoma recurrence compared to those on tacrolimus. In a group of 511 patients, recipients treated with everolimus exhibited a reduced risk of tumor recurrence after transplantation. This was particularly true for patients with more advanced tumors and who received the drug earlier and for longer periods. We recommend including everolimus in the post-transplant immunosuppressive regimen to optimize outcomes of liver transplantation for hepatocellular carcinoma.

**Abstract:**

To obtain long-term data on the use of everolimus in patients who underwent liver transplantation for hepatocellular carcinoma, we conducted a retrospective, single-center analysis of adult recipients transplanted between 2013 and 2021. Patients on everolimus-incorporating immunosuppression were matched with those on tacrolimus using an inverse probability of treatment weighting methodology. Two propensity-matched groups of patients were thus compared: 233 (45.6%) receiving everolimus versus 278 (54.4%) on tacrolimus. At a median (interquartile range) follow-up of 4.4 (3.8) years after transplantation, everolimus patients showed a reduced risk of recurrence versus tacrolimus (7.7% versus 16.9%; RR = 0.45; *p* = 0.002). At multivariable analysis, microvascular infiltration (HR = 1.22; *p* < 0.04) and a higher tumor grading (HR = 1.27; *p* < 0.04) were associated with higher recurrence rate while being within Milan criteria at transplant (HR = 0.56; *p* < 0.001), a successful pre-transplant downstaging (HR = 0.63; *p* = 0.01) and use of everolimus (HR = 0.46; *p* < 0.001) had a positive impact on the risk of post-transplant recurrence. EVR patients with earlier drug introduction (≤30 days; *p* < 0.001), longer treatment duration (*p* < 0.001), and higher drug exposure (≥5.9 ng/mL; *p* < 0.001) showed lower recurrence rates versus TAC. Based on our experience, everolimus provides a reduction in the relative risk of hepatocellular carcinoma recurrence, especially for advanced-stage patients and those with earlier drug administration, higher drug exposure, and longer time on treatment. These data advocate for early everolimus introduction after liver transplantation to reduce the attrition rate consequent to chronic immunosuppression.

## 1. Introduction

Complications of chronic immunosuppression impact long-term survival after liver transplantation (LT) [1]. Nearly all long-term survivors show diverse co-morbidities, the most frequent being hypertension, chronic kidney dysfunction (CKD), de novo malignancies, diabetes mellitus (DM), and metabolic disease [2]. Complications often arise from CNI use (e.g., cyclosporine (CyA) and tacrolimus (TAC)) alongside factors such as surgery, co-medication, age, and liver disease [1,2]. Introduced in clinical practice a decade ago [3,4,5], everolimus (EVR) is a member of the mammalian target of rapamycin (mTOR) inhibitors (mTORi) group together with sirolimus (SRL) and has been extensively studied in pre-clinical studies [6], registration trials [3,4,5], and real-life clinical practice [7]. Although approved for use with tacrolimus (TAC) [3], numerous reports have investigated its administration as a single immunosuppressant. This is especially true for treating CNI-associated nephrotoxicity and for the prevention/treatment of post-transplant de novo or recurrent malignancies [8]. Several pooled and meta-analyses have shown that incorporating EVR into immunosuppression therapy can improve renal function [9,10,11]. In addition, it can reduce the risk of post-transplant malignancies and recurrence of hepatocellular carcinoma (HCC) [12]. However, there is a large variability in reported outcomes across studies due to differences in patient selection, indications for EVR use, mode of administration and exposure, and study designs.

The antiproliferative profile of EVR makes it an ideal immunosuppressive agent for patients with HCC, especially for recipients with an advanced tumor stage or a higher risk of post-transplant recurrence [8]. However, the impact of any immunosuppressive strategy on the risk of post-transplant tumor recurrence is dependent on a complex interplay of tumor-related (i.e., stage at transplantation, response to pre-transplant downstaging, biological markers), condition-related (i.e., indication to transplantation, native liver disease, severity of liver dysfunction), and treatment-related correlates (i.e., time, mode, and duration of drug administration) [1]. To elucidate these interactions in large populations and in the long term, we conducted a retrospective analysis at our center on the post-transplant recurrence of HCC in adult LT recipients. 

## 2. Materials and Methods

### 2.1. Study Design 

We conducted a retrospective, single-center study at a National Health System (NHS)-based LT center.

### 2.2. Population

Two groups of adult HCC patients who underwent whole-size transplantation from deceased brain-dead donors (DCD) at our institution were compared using a two-tiered approach. Out of the entire cohort of patients transplanted at our center, we initially selected LT recipients with HCC on explant histology, excluding those with macrovascular tumor portal infiltration and mixed cellular lineage (i.e., hepato-cholangio-hepatocellular carcinoma (CHC-HCC)). We then compared patients who received immunosuppression with EVR (alone or in combination with tacrolimus (TAC)) to those who received TAC with or without mycophenolate derivatives (MPA) between 2013 and 2021 (Figure 1).

### 2.3. Primary Exposure

Our primary exposure was the use of EVR alone or in combination with TAC versus TAC (±MPA) in the post-transplant immunosuppressive schedule. 

### 2.4. Data Source

We used data from the regional transplant authority (CRT) and our institution’s recipient database, which we maintain prospectively. The CRT data system maintains records of all donors, candidates on the waitlist, and transplant recipients and ensures that the data are accurate, valid, and transparent. The local ethics committee of the University of Pisa approved all procedures (Prot. 0036349/2020).

### 2.5. Measure Outcomes

Our primary outcome was the cumulative incidence of HCC recurrence after transplantation in patients who received EVR versus those who received TAC. The secondary outcome was overall survival (OS). All these measures were treated as time-to-event occurrences. Data were censored at the time of the event, the last follow-up visit, or 31 December 2022, whichever came first.

### 2.6. Immunosuppression

During the study period, de novo immunosuppression consisted of triple or quadruple schedules with anti-CD25 (basiliximab, Simulect^®^, Novartis, Origgio (VA), Italy), calcineurin inhibitors (CNI), CyA or TAC, steroids (S), and antimetabolites (azathioprine (AZA) or mycophenolic acid (MPA) derivatives) according to era, pre-transplant, intra-operative, and post-transplant patient characteristics. 

When used for prophylaxis of HCC recurrence or prevention of TAC-related adverse events, EVR (Certican^®^, Novartis, Origgio (VA), Italy) was introduced ≥1 month after transplantation at a dose of 1.0 mg bid with antimetabolite elimination and stepwise TAC dose reduction, unless otherwise clinically indicated. Before EVR introduction, patients were tested for liver function (LFT), hematocrit, lipid profile, and creatinine/protein ratio. When used in combination schedules, EVR was adjusted to a trough level between 3 and 8 ng/mL and TAC between 3 and 5 ng/mL. When used alone, the EVR trough level was between 6 and 10 ng/mL, as clinically appropriate. When introduced for TAC-related complications, the time, mode, and dose of EVR were dependent on clinical indications. 

Rejection episodes were histologically confirmed and graded according to the BANFF classification system [13]. Treatment of rejection was with steroid boluses for non-HCV patients, whereas CNI and MPA dose increase/reintroduction were preferred for HCV-RNA positive recipients. In case of refractory rejection episodes, we tested the patients for donor-specific and anti-HLA antibodies and used plasmapheresis, intravenous immunoglobulin, or rabbit anti-thymocyte immunoglobulin (rATG) as clinically appropriate. Anti-CMV prophylaxis was used in recipients without acquired immunity (i.e., D+/R− and D−/R− combinations).

### 2.7. Drug Exposure Assay

EVR exposure was tested in whole blood with the QMS™ Everolimus Immunoassay, a homogeneous particle-enhanced turbidimetric immunoassay based on Quantitative Microparticle System (QMS^®^) technology. The mean EVR exposure was calculated based on samples obtained at our institution at follow-up visits. 

### 2.8. Special Considerations

The study period saw changes in the transplant procedure, perioperative management, and immunosuppressive schedules in accordance with technological advancements and scientific evidence. Donor-specific antibody (DSA) testing was usually performed on pre-transplant samples, but its positivity did not preclude transplantation. Celsior^®^ (IGL, Lissieu, France) was used for graft perfusion until 2017 and was replaced by Servator C^®^ (SALF, Cenate Sotto (BG), Italy). Bypassing or the classical technique were standard until 2017. Direct antiviral agents (DAAs) were used pre- or post-transplantation for HCV RNA-positive patients, as appropriate. In HBsAg-positive recipients, HBV recurrence prophylaxis was with lifelong anti-HBs immunoglobulin (HBIg) and antivirals, according to the era. Anti-HBsAg donors were used for anti-HBsAg recipients only after histological graft evaluation and exclusion of HDV co-infection. When anti-HBc positive grafts were used for naïve recipients, lifelong prophylaxis with antivirals was used. 

### 2.9. Pre-Transplant Management of HCC

The diagnosis of HCC was made according to the European Association for the Study of Liver Disease (EASL) guidelines [14]. Patients within the Barcelona Clinic Liver Cancer (BCLC) stage B (intermediate) were considered for transplantation [15]. Pre-transplant downstaging/bridging was indicated for patients with >3 cm tumor mass, those with AFP levels > 400 ng/mL, or with segmental portal infiltration. The down-staging technique was agreed upon during the tumor board case evaluation based on tumor size, location, and number of nodules. It consisted of transarterial chemoembolization (TACE), radiofrequency/microwave ablation, transarterial radioembolization (TARE), or surgery as appropriate. A successful downstaging procedure was associated with regression from beyond to within the Milan criteria as per pre-transplant imaging or explant histology. 

### 2.10. Post-Transplant Management of Recurrent HCC

HCC recurrence was diagnosed with a contrast-enhanced CT scan or MR based on the EASL guidelines [14]. Histology was performed when appropriate. These patients were converted to EVR monotherapy whenever feasible and referred for surgical, radiological treatment, or chemotherapy with VEGFR inhibitors based on multidisciplinary board consensus. 

### 2.11. Cut-Offs and Definitions

Cold ischemia time (CIT) was the time from donor cross-clamping until the organ was removed from the ice for implantation. The warm ischemia time (WIT) was the duration of ischemia during graft implantation. The definition of EAD was based on the research of Olthoff et al. [16]. Retrospectively, MELD scores were recalculated using the laboratory data available at the time of transplantation. Retrospectively, MELD scores were recalculated using the laboratory data available at the time of transplantation. The recurrence of HCV was confirmed through liver biopsy when HCV-RNA was positive. The recurrence of HBV infection was defined as the reappearance of HBsAg (±HBV DNA) in previously seroconverted patients, regardless of liver function. Renal function was evaluated using the Modification of Diet in Renal Disease-4 formula to estimate the glomerular filtration rate (eGFR). Chronic kidney dysfunction (CKD) was defined as follows: (a) estimated glomerular filtration rate (eGFR) <60 mL/min/1.73 m^2^ for a post-LT period greater than three months as per the Kidney Disease: Improving Global Outcomes (KDIGO) criteria in patients with previous eGFR ≥60 mL/min/1.73 m^2^ [17]; (b) evidence of intrinsic renal disease (proteinuria or kidney disease at ultrasound) [17]; or (c) presence of end-stage renal disease requiring renal replacement therapy [17]. Acute kidney injury (AKI) was defined as a doubling of baseline serum creatine (sCr) and/or a ≥50% reduction in eGFR within 14 days [17]. Deteriorating renal function was defined as ≥one-grade downward shift in the kidney function category according to the KDIGO classification system [18]. Post-transplant diabetes mellitus (PTDM) was determined using the comprehensive American Diabetes Association (ADA) 2018 criteria [19].

Arterial hypertension was defined as having a blood pressure of 140/90 mmHg or higher during two consecutive visits or requiring medication. Dyslipidemia was defined as having high levels of cholesterol (>220 mg/dL) and/or triglycerides (>200 mg/dL) at two consecutive visits. Biliary complications included biliary fistula, biliary stones, anastomotic biliary strictures, and posttransplant ischemic-type biliary lesion (ITBL), all of which were symptomatic and treated. ITBL was defined as any non-anastomotic stenosis that required an endoscopic or surgical procedure because of associated symptoms or signs without vascular complications.

### 2.12. Statistical Analyses

Initially, 2 groups of patients were extracted from the original population of HCC recipients transplanted at our center between 1996 and 2021 based on the presence of HCC in the explant histology and excluding those with mixed HCC–CHC and macrovascular portal infiltration in the explant liver. The EVR group included 463 patients who received the drug in their de novo immunosuppressive regimen for prophylaxis of post-transplant tumor recurrence or CNI-related complications other than HCC recurrence (i.e., renal function deterioration) and 556 patients who received CNI until complications or the latest follow-up (Figure 1). 

Later, we addressed the non-randomized design of the analysis by balancing the two groups using an inverse probability of treatment weighting (IPTW) approach. Two pseudo-groups were thus generated: EVR (i.e., patients on EVR ± TAC) and TAC. A propensity score for each patient in the original population was developed. The score was created using a multivariate logistic regression model that considered post-transplant HCC recurrence (no/yes) as the dependent variable. Eighteen confounding factors that have clinical significance were chosen to serve as covariates for both DFS and OS: patient sex [20]; age [21]; HCV [22]; diabetes mellitus at transplant [23]; CKD at transplant [24]; MELD score [25]; donor sex [26]; donor age [27]; cerebrovascular accident (CVA) as donor cause of death [28]; use of machine perfusion (MP) [29]; CIT [30]; pre-transplant tumor stage according to Milan criteria [31]; pre-transplant alpha-fetoprotein (AFP) [32]; efficacy of pre-transplant downstaging and defined as downstaging from beyond to within Milan criteria [33]; tumor stage at histology according to Milan criteria [31,32,33]; G3–G4 tumor grading [34]; presence of microvascular infiltration [34]; and mean TAC trough level within the first post-transplant month (≤10 ng/mL) [35].

To reduce artificial sample size modification in pseudo-data, stabilized weights (SW) were used with the following formula:*SW* = *p/PS for the study group and SW* = (*1 p*)/(*1 PS*) *for the control group*(1)
where p is the probability of etiology without considering covariates, and PS is the propensity score. A stabilized approach was preferred not to inflate the sample populations versus the original ones.

To avoid biases from population size, comparisons between covariate subgroups were reported as effect size (Cohen’s D value). Values with differences lower than |0.1| were considered negligible. Differences between |0.1| and |0.3| were small, those between |0.3| and |0.5| were moderate, and those above |0.5| were considerable.

After stabilizing IPTW, multivariate Cox regression analyses were performed to identify risk factors for HCC recurrence after transplantation. Hazard ratios (HR) and 95.0% confidence intervals (CI) were reported for significant variables. Survival analyses were performed using the Kaplan—Meier method, and the appropriate version of the log-rank test was adopted to compare the survival rates. Statistical significance was a *p*-value of 0.05.

Finally, after propensity matching, 3 further co-variates were used to explore the impact of EVR administration on the risk of HCC recurrence: (1) timing of EVR introduction, (2) duration of EVR treatment, and (3) EVR whole blood concentration throughout the study period. Patients were dichotomized according to median values, and sensitivity analyses were performed between recurring and non-recurring patients. 

Frequency and percentages, medians, interquartile ranges (IQR), or means and standard deviations (SD) were used to express variables as appropriate. Data errors and missingness were identified across the database and solved. Missing data in the dataset were addressed using a single imputation method. In detail, the median of nearby point imputation was adopted. In this case, the median was chosen as a measure of central tendency because the distribution of the managed variables was not symmetrical and was skewed toward one end. Recurrence-free survival (RFS) was defined as the time (months) from transplantation to diagnosis of HCC recurrence. OS was the time from transplantation to death or last observation. 

All statistical analyses and plots were performed using SPSS (version 27.0, SPSS Inc., Chicago, IL, USA). This study adheres to the ethical principles of the 1975 Declaration of Helsinki, as reflected in its a priori approval by the institution’s human research committee. It was also conducted according to the Guidelines for Strengthening the Reporting of Observational Studies in Epidemiology (STROBE).

## 3. Results 

### 3.1. Demographics and Clinical Characteristics of the Original Cohort

A total of 1019 adult patients transplanted for HCC at our center between 1996 and 2021 were initially considered (Table 1). Among them, 463 (45.4%) received EVR for HCC recurrence prophylaxis or complications other than HCC recurrence, whereas 556 (54.6%) received CNI (either TAC or CyA) as the primary de novo immunosuppressant until HCC recurrence. At a median (IQR) follow-up of 8.7 (8.1) years, 384 (37.6%) patients died, 41 (4.0%) were retransplanted, and 635 (62.3%) were alive. The 1, 5, and 10-year Kaplan–Meier probability (95% CI) of survival was 91% (89–93%) and 90.2% (88–93%), 77% (74–80%), and 75.8% (72–79%), 67% (63–71%), and 65% (61%–70.2%) for patient and graft, respectively.

Table 1 shows the clinical features of the recipients and donors from 1996 to 2021 and before matching. CNI patients were more frequently transplanted before 2013 (66.0% versus 31.7%; *p* < 0.0001), whereas EVR patients were more frequently transplanted beyond the Milan criteria (32.8% versus 18.1%; *p* < 0.0001), underwent more frequent pre-transplant downstaging procedures (59.6% versus 62.4%; *p* = 0.01), and showed higher median (IQR) AFP levels before surgery (46.3 (28) versus 4.7 (19) ng/mL; *p* = 0.002). Similarly, the proportion of G3–G4 grading (31.9% versus 25.1%; *p* = 0.01) and microvascular infiltration (39.5% versus 32.7%; *p* = 0.02) were higher in the EVR group.

### 3.2. Stabilized IPTW Effect 

To minimize the effect of selection biases caused by the non-randomized design of this retrospective study, the EVR and TAC pseudogroups were balanced using a stabilized IPTW method. Table 2 illustrates the results of the balancing procedure for the 18 potential confounders. Namely, seven variables showed insignificant differences before balancing, seven small, and four moderate. After IPTW, 14 variables showed insignificant differences, and three were small. The IPTW yielded two pseudogroups, i.e., 233 EVR patients versus 278 on TAC. 

### 3.3. Results in the Balanced Groups

After performing IPTW balancing, Table 3 presents the clinical characteristics of both groups. At a median follow-up of 4.4 (3.8) years after transplantation, the number of deaths, re-transplants, and HCC recurrences were 167 (32.7%), 20 (4.0%), and 65 (12.7%), respectively (Table 4). TAC patients showed higher death (37.8% versus 26.6%; RR = 1.41; *p* = 0.007) and HCC recurrence rates (16.9% versus 7.7%; RR = 2.2; *p* = 0.002). The main reasons for death in the TAC and EVR groups included HCC recurrence (15.1% versus 6.8%; RR = 2.51; *p* = 0.003), HCV recurrence (5.4% versus 6.9%; RR = 0.8; *p* = 0.48), infections/sepsis (5.7% versus 5.1%; RR = 1.11; *p* = 0.84), and de novo malignancies (3.9% versus 2.1%; RR = 1.85; *p* = 0.31) (Table 4). 

### 3.4. Re-Transplantation

A total of 20 (3.9%) patients were re-transplanted (Table 4). The main indication was primary non-function (PNF) of the liver graft, which accounted for 40% of such cases, followed by ischemic cholangiopathy (30%) and hepatic artery thrombosis (HAT) (15.0%). No significant difference was found in the causes of re-transplantation between the two groups.

### 3.5. HCC Recurrence

HCC recurred at a median (IQR) of 26.1 (48.7) months after transplantation and accounted for 15.1% of deaths in TAC patients versus 6.8% in EVR patients (RR = 2.51; *p* = 0.003). One-third of recurrences were in the liver only, whereas TAC patients showed more frequent multi-organ involvement (*p* = 0.002) (Table 4). In the EVR group, a numerically lower incidence of HCC recurrences was observed for patients on EVR monotherapy (n = 12, 6.8%) versus those on EVR + rTAC (n = 6, 10.8%; *p* = 0.33) was observed (Figure 1). 

### 3.6. Immunosuppression

Table 5 and Table 6 illustrate the immunosuppressive regimen in the EVR group with respect to indication, timing of introduction, duration of treatment, and median exposure. EVR was introduced at a median (IQR) post-transplantation interval of 30 (16) days for a median (IQR) of 46.6 (36.1) months. Median (IQR) EVR whole blood exposure was 5.8 (1.7) ng/mL. As many as 177 (75.9%) patients in the EVR group received EVR monotherapy, whereas 56 (24.1%) combined EVR and reduced-exposure TAC (Figure 1). At the latest follow-up, EVR was discontinued in 12 (5.1%) patients, for tBPAR in 4 (1.7%), progressing proteinuria in 5 (2.1%), and infection in 3 (1.3%) (Table 5).

In the EVR group, patients with HCC recurrence showed later EVR introduction (median (IQR) = 52 (26.4) versus 30 (12) days; *p* < 0.001), shorter duration of treatment (median (IQR) = 47.6 (57.0) versus 69.9 (24.8) months; *p* < 0.001), and lower drug exposure (median (IQR) = 3.65 (0.55) versus 5.9 (1.4) ng/mL; *p* < 0.001) (Table 6). 

### 3.7. Risk Factors for Recurrence-Free and Overall Survival

Table 7 illustrates the results of the multivariate analysis of risk factors for OS and RFS in the entire post-IPTW population. Successful pre-transplant downstaging (HR = 0.79; *p* = 0.006), being within Milan criteria at transplant (HR = 0.67; *p* < 0.001) and at histology (HR = 0.78; *p* = 0.02) and use of EVR (HR = 0.69; *p* = 0.009) had a positive impact on survival.

As for HCC recurrence, the presence of vascular micro-infiltration (HR = 1.22; *p* = 0.04) and higher tumor grading (HR = 1.27; *p* = 0.044) had a negative impact on RFS, while a successful pre-transplant downstaging (HR = 0.65; *p* = 0.01), being within Milan criteria at transplant (HR = 0.56; *p* = 0.01) and at histology (HR = 0.68; *p* = 0.012) and use of EVR (HR = 0.46; *p* < 0.001) had a positive impact on the probability of tumor-free survival. 

Figure 2: Kaplan–Meier (KM) survival function probability of post-transplant overall patient survival according to everolimus (EVR) versus tacrolimus (TAC) and Milan stage at transplantation. 1 = Milan-in EVR; 2 = Milan-in TAC; 3 = Milan-out EVR; 4 = Milan-out TAC. Overall log-rank *p* for the four groups was 0.03 (χ^2^ = 8.34; df = 3). 

Figure 3 illustrates the RFS according to Milan criteria at transplantation and the type of immunosuppressant (EVR versus TAC). The actuarial (95% CI) RFS for patients within the Milan criteria was 96% (94–100%), 87% (84–89%), and 82% (77–89%) at 1, 3, and 5 years in the EVR group versus 96% (CI 88–99%), 87% (74–91%), and 80% (72–84%) in the TAC group, respectively. On the other hand, it was 90% (82–94%), 78% (64–81%), and 68% (57–71%) at 1, 3, and 5 years for EVR patients exceeding the Milan criteria versus 86% (72–90%), 75% (66–84%), and 65% (52–69%) for TAC patients beyond the Milan criteria. (log-rank *p* = 0.006; χ^2^ = 12.13; df = 3 according to the Peto-Peto and Prentice). 

## 4. Discussion

This is one of the largest single-center series on the use of EVR in recipients of a liver graft for HCC, with long-term follow-up data available in the international literature. Despite the expansion of the practice of LT for advanced HCC, information on the impact of EVR or SRL is derived mainly from systematic reviews and meta-analyses incorporating both drugs [11,36,37,38,39,40], with only a few multi-institutional research studies [9,41,42,43] and real-life clinical series [44,45,46]. Although some authors dispute the advantage of mTORi for the reduction of post-transplant HCC recurrence [43], the overall evidence originating from these studies supports the use of mTORi for patients with HCC, earlier drug introduction and higher exposure levels to achieve more significant antiproliferative activity [46]. 

Dysregulation of the mTOR pathway is frequently observed in HCC [47]. However, transferring this evidence from the in vitro experimental setting to clinical practice is challenging because the overall impact of EVR on post-transplant HCC appears to be dependent on a complex interplay of molecular signaling, tumor cell viability, total tumor volume, plasma, and tumor drug concentration [48]. In addition, pre-transplant tumor characteristics, response to adjuvant treatment, type of native liver disease, quality of liver graft, surgery, and immunosuppressive regimens all account for the variable outcomes reported in the literature. One strategy to overcome these limitations is to analyze real-life clinical practice and large datasets using propensity-matching methodologies to reduce selection biases. Therefore, we opted for a retrospective comparison of two propensity-matched samples of HCC patients using the IPTW methodology. 

The analysis revealed that the use of EVR could reduce the risk of HCC recurrence and increase RFS and OS in LT recipients. This was especially true for tumors beyond the Milan criteria and for patients with earlier drug introduction (≤30 days), longer time on treatment (about 5 years), and higher median EVR exposure level (≥5.9 ng/mL). In such cases, a twofold reduction in the relative risk of HCC recurrence was observed alongside improved OS. In patients within the Milan criteria, the oncological impact of EVR-based immunosuppression was weaker, but the overarching benefits of EVR on renal function deterioration and de novo malignancies favor its use in this category of patients. 

Our data contradict previous experiences regarding the phase-3 trial on SRL reported by Geissler et al. in 2016 [41]. In their study, the authors compared 264 patients on SRL-free immunosuppression with 261 on SRL-incorporating regimens. Patients with low-risk hepatocellular carcinoma (within Milan criteria) experienced a significant benefit in RFS and OS within the first 3 to 5 years [41]. In our study, patients within the Milan criteria did not show different RFS and OS according to the type of immunosuppression, whereas the impact of EVR was observed for those beyond the Milan criteria (i.e., high-risk). To understand the apparent contradiction, the study conducted by Geissler et al. was a multicenter trial across 44 centers, with 42 centers situated in Europe, 1 in Australia, and another 1 in Canada. The study was conducted from 2006 (when the first patient visited) to 2014 (when the last patient finished their visit). This means their study was conducted well before our experience, from 2013 to 2021. Geissler’s patients were grouped based on the histology of their explant liver, and the low-risk and high-risk groups were determined using histologic data. In our study, the comparison between the treatment groups was based on the Milan stage as per the latest radiological examination before transplantation. Approximately 60% of the patients underwent downstaging/bridging procedures before transplantation, a quarter were successfully downstaged, and patients with macrovascular portal infiltration were excluded from the analysis. Their low AFP levels indicate that our patients beyond the Milan criteria were carefully selected and had less unfavorable biological criteria than those in Geissler’s study. Finally, our center has extensive experience managing EVR-related complications in LT recipients, as indicated by the lower EVR discontinuation and graft rejection rates in the current experience compared with previously reported data [3,4,5]. All these factors might explain the difference between our experience and Geissler and colleagues.

Transferring these considerations to current clinical practice, however, is not simple. 

Earlier EVR administration requires proper patient management due to a reported higher incidence of leukopenia, thrombocytopenia, dyslipidemia, and infections with the use of mTORi versus CNIs, as well as a higher discontinuation rate for patients on higher drug exposure [3,10]. Therefore, the initially recommended exposure levels for EVR monotherapy (i.e., 6–10 ng/mL) [3,4,5] are seldom implemented in clinical practice to avoid concentration-related adverse events. Furthermore, the current focus on immunosuppression minimization has expanded from CNI to include antimetabolites and mTORi alongside increased concerns about transplant recipients’ quality of life [49]. 

Despite its large number of patients and robust statistical methodology, our study has several limitations. First, retrospective designs do not always provide sufficient detailed information for clinically transferable data. For instance, we could not appropriately investigate the role of DSA positivity on post-transplant outcomes [50], the role of inflammatory markers (beyond the NLR), or the role of serologic donor–recipient mismatching regarding HBV prophylaxis. Pre-transplant information on tumor clinical course and biology was also limited, such as the availability of drug exposure levels for patients with longer follow-ups. Furthermore, this study was biased by the initial experience with EVR at our center and the learning curve effect in terms of patient selection, drug discontinuation rates, and post-transplant therapeutic strategies for HCC recurrence.

Second, we excluded patients with unfavorable tumor characteristics from the current analysis, i.e., neoplastic portal thrombosis, because of its negative impact on RFS and OS and the limited number of cases at our center. Given the clinical expansion of the Milan criteria and the introduction of novel neoadjuvant pre-transplant treatments (i.e., immunotherapy), it would be interesting to analyze the relative benefit of EVR-incorporating immunosuppression for this category of high-risk patients. 

## 5. Conclusions

In conclusion, based on our results, EVR allows us to mitigate the risk of post-transplant HCC recurrence, especially for tumors beyond the Milan criteria and patients with earlier drug introduction, higher drug exposure, and longer treatment duration. These data advocate the use of mTORi for patients with HCC to reduce the attrition rate of chronic immunosuppression after LT.

## Figures and Tables

**Figure 1 cancers-16-01243-f001:**
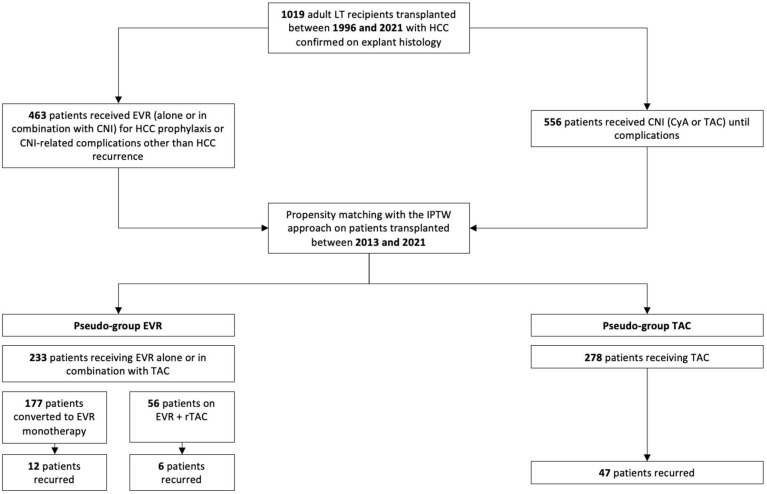
Flowchart of the study cohort.

**Figure 2 cancers-16-01243-f002:**
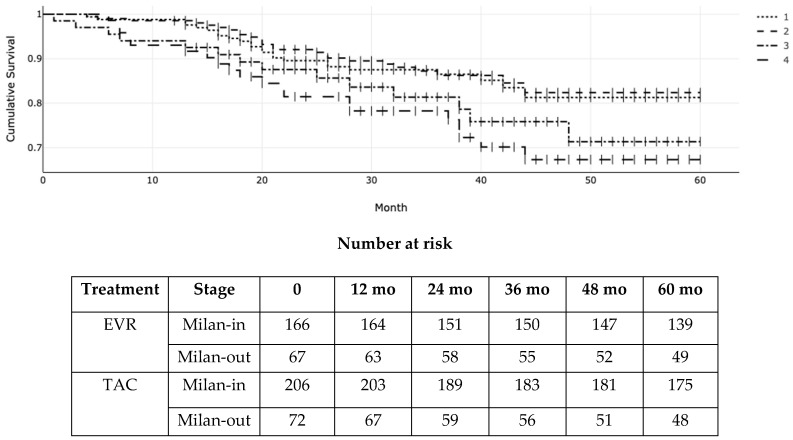
Illustrates the OS according to the Milan criteria at transplant and the type of immunosuppressant (EVR versus TAC). The 1-, 3-, and 5-year actuarial OS (95% CI) for EVR patients within the Milan criteria as per pre-transplant radiologic staging were 98% (93–99%), 89% (82–90%), and 84% (79–88%) versus 98% (94–99%), 89% (83–90%), and 85% (80–87%) for EVR and TAC, respectively. For patients exceeding the Milan criteria, the OS (95% CI) was 94% (91–96%), 84% (81–86%), and 76% (72–79%) versus 92% (89–94%), 79% (73–80%), and 69% (66–70%) for EVR and TAC, respectively. Due to the crossing of the Kaplan–Meier curves of patients within the Milan criteria indicating non-proportional hazards, we used Peto-Peto-Prentice’s version of the log-rank test to measure differences across patient groups (log-rank *p* = 0.03; χ^2^ = 8.34; df = 3).

**Figure 3 cancers-16-01243-f003:**
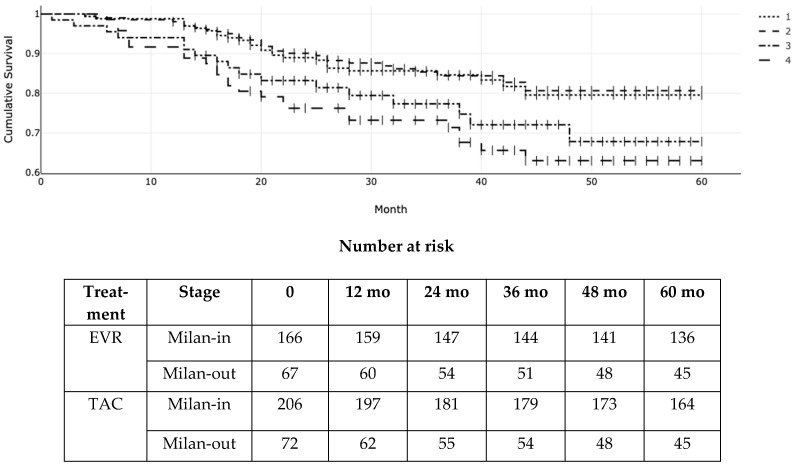
Kaplan–Meier (KM) probability of post-transplant recurrence-free survival (RFS) according to everolimus (EVR) versus tacrolimus (TAC) and Milan stage at transplantation. 1 = Milan-in EVR; 2 = Milan-in TAC; 3 = Milan-out EVR; 4 = Milan-out TAC. Log-rank *p* = 0.006 (χ^2^ = 12.13; df = 3).

**Table 1 cancers-16-01243-t001:** Demographic and clinical characteristics of interest of the population of patients with HCC transplanted between 1996 and 2021 at our center.

Variable	EVR (#463)	CNI (#556)	*p*
**Recipient**			
Male sex, n (%)	386 (83.4)	487 (87.6)	0.55
Age at transplant (median, IQR), years	56 (10)	56 (10)	0.28
Indication to transplant, n (%)			
HCV	237 (55.5)	294 (52.8)	0.59
HBV (±HDV)	123 (26.5)	153 (27.5)	0.73
HCV-HBV(±HDV)	17 (3.7)	20 (3.5)	0.94
Alcohol	54 (11.6)	59 (9.5)	0.59
NAFLD	20 (4.3)	24 (4.3)	0.99
Autoimmune/PSC	12 (5.2)	6 (1.1)	0.67
Lab-MELD at transplant (median, IQR)	7 (6)	8 (7)	0.45
DM at transplant, n (%)	113 (24.4)	144 (25.8)	0.58
CKD at transplant, n (%)	27 (5.8)	39 (7.0)	0.44
Hypertension at transplant, n (%)	69 (14.9)	76 (13.6)	0.57
<2013, n (%)	147 (31.7)	387 (66.0)	<0.0001
TAC, n (%)	403 (86.8)	313 (56.3)	<0.0001
Mean TAC exposure >10 ng/mL within the first month post-transplantation	127 (27.4)	172 (31.0)	0.22
**Donor**			
Male sex, n (%)	241 (52.0)	281 (50.5)	0.63
Age, median (IQR)	69 (25)	67 (26)	0.78
ICU stay, median (IQR) days	3 (4)	3 (4)	0.67
CVA as cause of death, n (%)	333 (71.9)	411 (73.9)	0.47
Anti-HCV-positive, n (%)	4 (0.86)	0 (0)	0.58
Anti-HBc-positive, n (%)	60 (12.9)	77 (13.8)	0.67
Cardiac arrest episodes, n (%)	43 (9.3)	42 (7.5)	0.31
Use of inotropes, n (%)	407 (87.9)	483 (86.8)	0.62
**HCC**			
Tumor nodules *, median (IQR)	2 (1)	2 (1)	0.78
Largest nodule size *, median (IQR) (mm)	28 (18)	25 (15)	0.04
Total tumor size *, median (IQR) (mm)	39.5 (25)	36.5 (36)	0.003
Exceeding Milan criteria at transplant *, n (%)	152 (32.8)	101 (18.1)	<0.0001
Pre-transplant treatment, n (%)			
None, n (%)	141 (30.4)	209 (37.6)	0.01
TACE, n (%)	229 (49.4)	307 (55.2)	0.06
RFA/MW, n (%)	33 (7.1)	22 (3.9)	0.02
PEI, n (%)	6 (1.3)	12 (2.1)	0.29
Resection, n (%)	6 (1.3)	4 (0.7)	0.35
TACE + RFA/MW, n (%)	42 (9.1)	2 (0.5)	<0.0001
TARE, n (%)	6 (1.3)	0 (0)	0.008
Successful downstaging **, n (%)	75 (16.2)	45 (8.1)	0.0006
AFP at transplant, median (IQR) (ng/mL)	46.3 (28)	4.7 (19)	0.002
Milan-out at explant histology, n (%)	120 (25.9)	167 (30.0)	0.98
G3-4, n (%)	148 (31.9)	140 (25.1)	0.01
Microvascular infiltration, n (%)	88 (39.5)	182 (32.7)	0.02
**Transplantation**			
CIT, median (IQR) (min)	424 (89)	420 (101)	0.09
MP, n (%)	9 (1.9)	7 (1.2)	0.89
Re-transplantation, n (%)	18 (3.8)	23 (4.1)	0.45
B cell and/or T cell positive X-match, n (%)	35 (7.5)	48 (8.6)	0.53
NLR, median (IQR)	2.2 (0.2)	2.1 (0.3)	0.68

NOTE: AFP, alpha-fetoprotein; CKD, chronic kidney failure; CNI, calcineurin inhibitor; DM, diabetes mellitus; EVR, everolimus; HBV, hepatitis B virus; HCV, hepatitis C virus; HDV, hepatitis delta virus; IQR, interquartile range; MELD, model for end-stage liver disease; NAFLD, non-alcoholic fatty liver disease; MW, microwave ablation; MP, machine perfusion; NLR, neutrophil-to-lymphocyte ratio; PEI, percutaneous ethanol injection; PSC, primary sclerosing cholangitis; RFA, radiofrequency ablation; TAC, tacrolimus; TACE, trans-arterial chemoembolization; TARE, trans-arterial radioembolization. * Radiological; ** Radiological, as downstaged from outside to within Milan criteria.

**Table 2 cancers-16-01243-t002:** Effect of stabilized IPTW in the population on the variables used for balancing the two groups.

Variables	Pre-IPTW			Post-IPTW		
	EVR (n = 463)	CNI (n = 556)	Cohen’s D-Value	EVR (n = 233)	TAC (n = 278)	Cohen’s D-Value
	Mean (±SD)			Mean (±SD)		
Patient male sex	0.83 ± 0.15	0.87 ± 0.14	0.05	0.81 ± 0.17	0.82 ± 0.15	0.05
Patient age, years	55.9 ± 3.92	56.4 ± 3.46	−0.20	55.1 ± 0.55	55.3 ± 0.53	−0.03
HCV	55.5 ± 0.70	55.8 ± 0.58	−0.42	24.3 ± 0.56	24.1 ± 0.52	0.01
Patient diabetes	0.24 ± 0.50	0.26 ± 0.45	0.12	0.23 ± 0.50	0.24 ± 0.50	0.00
Patient CKD	0.05 ± 0.02	0.07 ± 0.42	0.42	0.05 ± 0.01	0.05 ± 0.01	0.01
MELD	0.07 ± 0.26	0.11 ± 0.33	−0.15	0.08 ± 0.38	0.07 ± 0.37	0.01
Donor male sex	0.52 ± 0.38	0.50 ± 0.41	−0.08	0.51 ± 0.28	0.50 ± 0.28	0.01
Donor age, years	0.69 ± 0.65	0.67 ± 0.64	0.01	0.68 ± 0.38	0.67 ± 0.37	0.03
Donor cause of death (CVA)	0.71 ± 0.50	0.73 ± 0.40	0.13	0.71 ± 0.46	0.72 ± 0.45	−0.02
MP	0.09 ± 0.02	0.08 ± 0.02	0.01	0.08 ± 0.02	0.08 ± 0.02	0.00
CIT, minutes	431.77 ± 79.02	423.50 ± 85.79	0.10	0.53 ± 0.50	0.53 ± 0.50	0.01
Milan-out stage, radiologic	32.8 ± 2.33	19 ± 1.65	0.42	29.2 ± 1.2	26.3 ± 0.9	0.02
Tumor downstaging	0.17 ± 0.05	0.08 ± 0.04	0.43	0.10 ± 0.02	0.08 ± 0.08	0.04
AFP at transplant	0.45 ± 0.27	0.04 ± 0.02	0.38	0.23 ± 0.04	0.19 ± 0.06	0.14
Milan-out stage, histology	26.1 ± 1.5	29.1 ± 2.3	0.16	24.0 ± 2.3	22.0 ± 2.8	0.12
G3–G4	32.0 ± 2.4	25 ± 1.8	0.23	29.0 ± 2.3	28.2 ± 2.0	0.21
Microinfiltration	40.2 ± 1.2	33.7 ± 1.7	0.26	38.1 ± 2.3	32.5 ± 1.9	0.20
Mean TAC exposure > 10 ng/mL within the first month	0.28 ± 0.04	0.32 ± 0.45	0.13	0.15 ± 0.02	0.13 ± 0.03	0.00

NOTE: AFP, alpha-fetoprotein; CIT, cold ischemia time; CKD, chronic kidney disease; CVA, cerebrovascular accident; G, grading; HCV, hepatitis C virus; IPTW, inverse probability therapy weighting; MP, machine perfusion; n, number; SD, standard deviation; TAC, tacrolimus.

**Table 3 cancers-16-01243-t003:** The characteristics of the two pseudo-groups after IPTW matching.

Variable	EVR (#233)	TAC (#278)	*p*
Recipient			
Male sex, n (%)	192 (82.4)	228 (82.0)	0.90
Age at transplant (median, IQR), years	55.5 (9)	55.3 (10)	0.89
HCV, n (%)	58 (24.3)	69 (24.8)	1
Lab-MELD at transplant (median, IQR) *	8 (6)	7 (7)	0.78
DM at transplant, n (%)	53 (22.7)	66 (23.7)	0.83
CKD at transplant, n (%)	12 (5.1)	14 (5.0)	1
Mean TAC exposure >10 ng/mL within the first-month post-transplantation	35 (15.0)	36 (12.9)	0.52
Donor			
Male sex, n (%)	118 (50.6)	140 (50.3)	1
Age, median (IQR)	68.0 (23)	67 (26)	0.89
CVA as cause of death, n (%)	181 (77.7)	200 (71.9)	0.15
HCC			
Exceeding Milan criteria at transplant *, n (%)	67 (28.7)	72 (25.8)	0.48
Successful downstaging **, n (%)	24 (10.3)	23 (8.2)	0.44
AFP at transplant, median (IQR) (ng/mL)	23.3 (18)	19 (11)	0.56
Milan-out at explant histology, n (%)	55 (23.6)	62 (22.3)	0.75
G3–4, n (%)	67 (28.7)	78 (28.1)	0.92
Microvascular infiltration, n (%)	88 (37.8)	91 (32.7)	0.26
Transplantation			
CIT, median (IQR) (min)	432 (89)	489 (101)	0.06
MP, n (%)	9 (1.9)	7 (1.2)	0.89
B cell and/or T cell positive X-match, n (%)	16 (6.9)	21 (7.5)	0.76
NLR, median (IQR)	2.3 (0.2)	2.1 (0.2)	0.89

NOTE: AFP, alpha-fetoprotein; CIT, cold ischemia time; CKD, chronic kidney failure; CNI, calcineurin inhibitor; DM, diabetes mellitus; EVR, everolimus; HCV, hepatitis C virus; IPTW, inverse probability of treatment weighting; IQR, interquartile range; MELD, model for end-stage liver disease; MP, machine perfusion; TAC, tacrolimus. * Radiological; ** Radiological, as downstaged from outside to within Milan criteria.

**Table 4 cancers-16-01243-t004:** Results in the study groups after stabilized IPTW matching.

Variable	EVR (#233)	TAC (#278)	*p*
Death, n (%)	62 (26.6)	105 (37.8)	0.007
HCC recurrence, n (%)	16 (6.8)	42 (15.1)	0.003
HCV recurrence, n (%)	16 (6.9)	15 (5.4)	0.48
Incomplete/delayed graft function, n (%)	1 (0.4)	2 (0.7)	0.22
MACE, n (%)	2 (0.8)	6 (2.1)	0.30
Intra/peri-operative, n (%)	2 (0.8)	3 (1.1)	1
Ischemic cholangiopathy, n (%)	4 (1.7)	7 (2.5)	0.76
Infection/sepsis, n (%)	12 (5.1)	16 (5.7)	0.84
De novo malignancy, n (%)	5 (2.1)	11 (3.9)	0.31
Stroke, n (%)	4 (1.7)	3 (1.1)	0.70
Re-transplantation, n (%)	9 (3.9)	11 (3.9)	1
Ischemic cholangiopathy, n (%)	3 (1.3)	3 (1.1)	1
PNF, n (%)	3 (1.3)	5 (1.8)	0.73
HAT, n %	2 (0.8)	1 (0.3)	0.59
Chronic rejection, n (%)	1 (0.4)	1 (0.3)	1
HCV recurrence, n (%)	0 (0)	1 (0.3)	0.99
HCC recurrence, n (%)	18 (7.7)	47 (16.9)	0.002
Liver only, n (%)	7 (3.0)	15 (5.4)	0.19
Liver and lung, n (%)	1 (0.4)	8 (2.8)	0.04
Liver and bone, n (%)	0 (0)	4 (1.4)	0.12
Lung only, n (%)	4 (1.7)	9 (1.8)	0.39
Bone only, n (%)	4 (1.7)	1 (0.2)	0.18
Lung and bone, n (%)	1 (0.4)	4 (1.4)	0.38
Nodes, n (%)	1 (0.4)	6 (2.1)	0.13
>2 organs, n (%)	2 (0.8)	16 (5.7)	0.002

NOTE: EVR, everolimus; HAT, hepatic artery thrombosis; HCC, hepatocellular carcinoma; HCV, hepatitis C virus; IPTW, inverse probability of treatment weighting; IQR, interquartile range; MACE, major cardiovascular events; PNF, primary non-function; TAC, tacrolimus.

**Table 5 cancers-16-01243-t005:** Immunosuppression in the EVR study groups after stabilized IPTW matching.

Variable	EVR (#233)
Reason for EVR use, n (%)	
HCC recurrence prophylaxis, n (%)	212 (91.0)
Deteriorating renal function, n (%)	14 (6.0)
Neurologic complication, n (%)	4 (1.7)
MACE, n (%)	3 (1.2)
Timing of EVR introduction, median (IQR) (days)	30 (16)
Duration of EVR treatment, median (IQR) (months)	46.6 (36.1)
EVR whole-blood exposure, median (IQR) (ng/mL)	5.8 (1.7)
EVR monotherapy, n (%)	177 (75.9)
EVR ± TAC, n (%)	56 (24.1)
EVR discontinuation, n (%)	12 (5.1)
t/BPAR, n (%)	4 (1.7)
Progressing proteinuria, n (%)	5 (2.1)
Infection, n (%)	3 (1.3)

NOTE: EVR, everolimus; HCC, hepatocellular carcinoma; IPTW, inverse probability of treatment weighting; IQR, interquartile range; MACE, major cardiovascular events.

**Table 6 cancers-16-01243-t006:** EVR mode of administration and exposure in recurring versus non-recurring patients of the EVR group.

Variable	Recurring HCC (#18)	Non-Recurring HCC (#215)	*p*
Timing of EVR introduction, median (IQR) (days)	52 (26.4)	30 (12)	<0.001
Duration of EVR treatment, median (IQR) (months)	46.5 (57.0)	69.9 (24.8)	<0.001
EVR whole-blood exposure, median (IQR) (ng/mL)	3.65 (0.55)	5.9 (1.4)	<0.001

NOTE: EVR, everolimus; HCC, hepatocellular carcinoma; IQR, interquartile range.

**Table 7 cancers-16-01243-t007:** Results of the multivariable analysis of risk factors for both recurrence-free and overall survival.

Variable	Coefficients (95%CI)	SE	z	HR	*p*
OS					
Successful pre-transplant downstaging	0.6 (0.15; 1.06)	0.23	2.6	0.79	0.006
Within Milan criteria at transplant	−1.15 (−1.61; −0.7)	0.23	5.02	0.67	<0.01
Within Milan criteria at histology	0.01 (0; 0.01)	0	2.41	0.78	0.02
Micro-infiltration	0.39 (−0.01; 0.78)	0.2	1.91	1.13	0.056
G3-G4	0.25 (0.01; 0.5)	0.12	2.02	1.18	0.077
EVR	−0.59 (−1.02; −0.16)	0.22	2.7	0.69	0.009
RFS					
Successful pre-transplant downstaging	0.57 (0.12; 1.02)	0.23	2.47	0.65	0.01
Within Milan criteria at transplant	−1.18 (−1.63; −0.72)	0.23	5.11	0.56	0.01
Within Milan criteria at histology	0.01 (0; 0.01)	0	2.52	0.68	0.012
Micro-infiltration	0.42 (0.02; 0.81)	0.2	2.06	1.22	0.04
G3-G4	0.22 (−0.02; 0.47)	0.13	1.77	1.27	0.04
EVR	−0.78 (1.2; −0.36)	0.21	3.66	0.46	<0.001

NOTE: AFP, alpha-fetoprotein; EVR, everolimus; HCV, hepatitis C virus; OS, overall survival; RFS, recurrence-free survival.

## Data Availability

The data can be shared up on reasonable request.

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
