# Peer review of "Everolimus Mitigates the Risk of Hepatocellular Carcinoma Recurrence after Liver Transplantation"

_cancers, 2024, doi:10.3390/cancers16071243_

Round 1

Reviewer 1 Report

Comments and Suggestions for Authors

Dear Authors,

I would like to appreciate your efforts for the manuscript entitled "Everolimus mitigates the risk of hepatocellular carcinoma recurrence after liver transplantation "

I have some comments that I wish it may help:

1- Since HLA matching (MHC class I and II) between the donor ( which is DCD) and the recipient is not necessary for many clinical consideration but it is favorable as well, and Have you ever check for Donor specific antibodies (DSA) records for donors?

As an example, here are many papers manifesting the importance DSA in clinical decision in liver transplantation?

https://pubmed.ncbi.nlm.nih.gov/31810796/

https://pubmed.ncbi.nlm.nih.gov/31810796/

https://bmcgastroenterol.biomedcentral.com/articles/10.1186/s12876-020-01427-4

https://www.mdpi.com/2077-0383/9/12/3986

2- For the hypothesis that virus like HCV and HBc may contribute to the immune reactivation or unfavorable condition in infected patients, have you checked for or measured the viral titration after transplantation and during the  EVR and TAC therapy ?

3- I think if you correlate the clinical data with hemato-immunological data, would be strong to assess the immunosuppression efficiency in addition to the measured data.

Thank you.

Author Response

REVIEWER 1

RE: Since HLA matching (MHC class I and II) between the donor (which is DCD) and the recipient is not necessary for many clinical considerations, but it is favorable as well, and Have you ever checked for Donor-specific antibodies (DSA) records for donors? As an example, here are many papers manifesting the importance of DSA in clinical decisions in liver transplantation.

https://pubmed.ncbi.nlm.nih.gov/31810796/

https://pubmed.ncbi.nlm.nih.gov/31810796/

https://bmcgastroenterol.biomedcentral.com/articles/10.1186/s12876-020-01427-4

https://www.mdpi.com/2077-0383/9/12/3986

AU: Thank you for your comments. DSA is tested on pre-transplant tubes at our center but not urgently; the results are available post-operatively. In simpler terms, patients are transplanted without knowing their DSA titers. This is due to the controversial importance of a positive x-match in liver transplantation, although, as correctly stated, several reports highlight the role of DSA in the clinical outcome. We have added this to the text in the Materials and Methods: “Donor-specific antibody (DSA) testing was usually performed on pre-transplant samples, but its positivity did not preclude transplantation”.

We also reviewed the present database and introduced data on DSA. The presence of a positive B cell/T cell X-match in our cohort was added as a variable to the Table. Unfortunately, a positive X match was not associated with a higher/lower incidence of post-transplant HCC recurrence in our experience. One of the reasons might be the expanded use of anti-IL2R at our institution (speculative). We have commented on this in the text. “For instance, we could not appropriately investigate the role of DSA positivity on post-transplant outcomes (51), the role of inflammatory markers (beyond the NLR), or the role of serologic donor-recipient mismatching regarding HBV prophylaxis (52)”.

2- RE: For the hypothesis that virus like HCV and HBc may contribute to the immune reactivation or unfavorable condition in infected patients, have you checked for or measured the viral titration after transplantation and during the  EVR and TAC therapy ?

AU: Thank you for your comments. The present cohort includes patients transplanted between 2013 and 2021 and matched on propensity scores according to the IPTW approach. Since 2014, HCV RNA-positive patients at transplantation have undergone post-transplant antiviral treatment with available anti-HCV drugs (sofosbuvir, ledipasvir, velpatasvir, etc.…).

Regarding anti-HBc recipients, patients receiving anti-HBc grafts undergo recurrence prophylaxis with lamivudine, while patients receiving naïve liver grafts are not administered prophylaxis, as per international guidelines. A national study on anti-HBs titers in HBsAg-positive liver transplant recipients on post-transplant anti-HBV prophylaxis (immunoglobulin) is ongoing nationwide. Unfortunately, the current study did not explore the role of HBV prophylaxis on HCC recurrence. We have clarified this in the text in the Materials and Methods: “Direct antiviral agents (DAAs) were used pre or post-transplantation for HCV RNA positive patients, as appropriate. In HBsAg positive recipients, HBV recurrence prophylaxis was with lifelong HBIg and antivirals, according to era. Anti-HBsAg donors were used for anti-HBsAg recipients only, after histologic graft evaluation and exclusion of HDV co-infection. When anti-HBc positive grafts were used for naïve recipients, lifelong prophylaxis with antivirals was used”.  

RE: I think if you correlate the clinical data with hemato-immunological data, would be strong to assess the immunosuppression efficiency in addition to the measured data.

AU: thank you for your comment. We do agree on the predictive role of hemato-immunological data. Due to the retrospective design of the current study, the only variable we could integrate into the paper was the neutrophil-to-leukocyte (NLR) ratio. This was available for all transplant recipients since it is routinely performed at transplantation. The variable was distributed consistently across the 2 groups, with only numerical difference (TAC patients had higher values than EVR patients). However, in EVR-based immunosuppression, the role of inflammatory markers is a topic that deserves further investigation due to the leukopenic and thrombocytopenic effects of the drug. This is worth investigating in future studies.

Thank you.

Reviewer 2 Report

Comments and Suggestions for Authors

De Simone and colleagues conducted a retrospective, single-center analysis of adult recipients transplanted between 2013 and 2021 in order to obtain long-term data on the use of everolimus in patients who underwent liver transplantation for hepatocellular carcinoma.

Major points:

1.     In the method section you describe the primary exposure to be EVR alone or in combination with TAC versus TAC. Therefore, it would be interesting to know the outcome of the subgroup EVR alone

2.     In abstract and methods, it is indicated that patients were included between 2013 and 2021. However, in the statistical analysis it is mentioned that patients between 1996 and 2021 were included as also in Table 1. This is somewhat confusing. Could you please specify and provide a flowchart about the groups.

3.     In the result section 3.2. you describe the IPTW groups. However, the description of the confounders do not match with the Cohen’s D value described in the methods and table 2. For instance, you describe 1 considerable variable before matching but there is no value greater than 0.5!

4.     In figure 1 and 2. How can the significance between Milan out EVR and TAC be the same as between Milan in EVR and Milan out TAC? What kind of test was used to run the subgroups? Why does the line of Milan in EVR and TAC crosses twice. These results should be checked by a proper statistician!

5.     The major finding of the study that “everolimus provides a reduction of the relative risk of hepatocellular carcinoma recurrence, especially for advanced-stage patients” is in the opposite to data of the SiLVER Study “RFS and OS benefit is evident in the first 3 to 5 years, especially in low-risk patients” (Transplantation 2016;100: 116 –125). However, you do not comment at all on this in your discussion! How do you explain this discrepancy? From a mole 
cancers-2845743 cular biology stand point it would be indeed make more sense that the low risk patients with low tumor burden benefit more!

Minor points:

1.     The subheading in table 3 are not in line (eg. DONOR is in the middle and HCC on the left)

2.     The p value in table 4 are not in the corresponding line.

Author Response

We thank the reviewer for the suggestions.

RE: In the method section you describe the primary exposure to be EVR alone or in combination with TAC versus TAC. Therefore, it would be interesting to know the outcome of the subgroup EVR alone.

AU: Thank you for your comment. We have added this information to the text, Table 5, and Figure 1. In the EVR group, 177 patients were on EVR monotherapy, 56 on combined immunosuppressants. We have also added the outcome (i.e., HCC recurrence for the group of EVR-alone patients: this was 6/56 versus 12/177, which was not significant.

RE: In abstract and methods, it is indicated that patients were included between 2013 and 2021. However, in the statistical analysis it is mentioned that patients between 1996 and 2021 were included as also in Table 1. This is somewhat confusing. Could you please specify and provide a flowchart about the groups.

AU: Thank you for your valuable insight. We have added a flowchart to clarify this (Figure 1).

RE: In the result section 3.2. you describe the IPTW groups. However, the description of the confounders do not match with the Cohen’s D value described in the methods and table 2. For instance, you describe 1 considerable variable before matching but there is no value greater than 0.5!

  1. Thank you for your comment. You are right. We have aligned the text to the Table.

RE: In figure 1 and 2. How can the significance between Milan out EVR and TAC be the same as between Milan in EVR and Milan out TAC? What kind of test was used to run the subgroups? Why does the line of Milan in EVR and TAC crosses twice. These results should be checked by a proper statistician!

AU: thank you for your comments. As you correctly pointed out, there is some K-M curves crossing for patients within the Milan criteria on EVR and TAC. We have checked this with the help of a statistician. The survival crossing means that the risk of death and HCC was nonproportional. Therefore, we performed the log-rank analysis using the Peto-Peto and Prentice version of the test, which accounts for a higher incidence of events (death and recurrences) during the early post-transplant phases. We have added this to the text and formatted the Figures accordingly. Thank you for your valuable and informative remarks.

RE: The major finding of the study that “everolimus provides a reduction of the relative risk of hepatocellular carcinoma recurrence, especially for advanced-stage patients” is in the opposite to data of the SiLVER Study “RFS and OS benefit is evident in the first 3 to 5 years, especially in low-risk patients” (Transplantation 2016;100: 116 –125). However, you do not comment at all on this in your discussion! How do you explain this discrepancy? From a mole cancers-2845743 cular biology stand point it would be indeed make more sense that the low risk patients with low tumor burden benefit more!

AU: Thank you for your comments. This is true and we have added some explanations in the Discussion. Thera are varied reasons to explain this, but we do believe that our Milan out patients were less unfavourable than the ones in the SILVER study.

“Our data contradicts previous experiences regarding the phase-3 trial on SRL reported by Geissler et al in 2016 (49). In their study, the authors compared 264 patients on SRL-free immunosuppression with 261 on SRL-incorporating regimens. They observed that a RFS and OS benefit was evident in the first 3 to 5 years, especially in low-risk patients (i.e., within Milan criteria) (49). In our study, patients within the Milan criteria did not show different RFS and OS according to type of immunosuppression, while the impact of EVR was observed for those beyond the Milan criteria (i.e., high-risk). In order to understand the apparent contradiction, it is important to note that the study conducted by Geissler and colleagues was a multicenter trial across 44 centers, with 42 centers situated in Europe, 1 in Australia, and another 1 in Canada. The study was conducted from 2006 (when the first patient visited) to 2014 (when the last patient finished their visit). This means that their study was carried out well before our experience, which took place from 2013 to 2021. Geissler's patients were grouped based on the histology of their explant liver, and the low-risk and high-risk groups were determined using histologic data. In our study, the comparison between the treatment groups was based on the Milan stage as per the latest radiologic examination before transplantation. About 60% of the patients underwent downstaging/bridging procedures before transplantation, a quarter was successfully downstaged, and patients with macrovascular portal infiltration were excluded from the analysis. Their low AFP levels indicate that our patients beyond the Milan criteria were carefully selected and had less unfavorable biological criteria than the ones in Geissler’s study. Finally, our center has extensive experience managing EVR-related complications in LT recipients, as indicated by the lower EVR discontinuation and graft rejection rates rate in the current experience compared with previously reported data (3-5). All these factors might contribute to explaining the difference between our experience and the one by Geissler and colleagues”.

Minor points:

  1. The subheading in table 3 are not in line (eg. DONOR is in the middle and HCC on the left)
  2. The p value in table 4 are not in the corresponding line.

AU: We have realigned all this and will check for the final proofs, if the paper is accepted.

Round 2

Reviewer 1 Report

Comments and Suggestions for Authors

Dear Authors,

Thank you for your updated version of the manuscript. It becomes clearer and covers more aspects.

However, plagiarism is high (28%). Could you please provide with an updated version of lower plagiarism (less than 20%).

Good Luck.

Author Response

Dear reviewer,

Thank you for your comments. We have amended the text as requested and checked for plagiarism through ENAGO. Please, be informed that about 3% of plagiarism originates from the initial version of the paper being published before peer review (on preprints). 

Based on the analysis (see attached file) the text similarity rate is 18%.

Cheers

Paolo De Simone

Reviewer 2 Report

Comments and Suggestions for Authors

Main concerns have been addressed and manuscript is improved.

Author Response

Thank you for your help

Cheers

Paolo De Simone

Round 3

Reviewer 1 Report

Comments and Suggestions for Authors

Dear Authors,

Thank you for your improvements in the manuscript.

I have no comments except the plagiarism level which is mostly comes from your reprint, which is acceptable.

Good luck